# RE-Adapt: Reverse Engineered Adaptation of Large Language Models

## Abstract

We introduce RE-Adapt, an approach to fine-tuning large language models on new domains without degrading any pre-existing instruction-tuning. We reverse engineer an adapter which isolates what an instruction-tuned model has learned beyond its corresponding pretrained base model. Importantly, this requires no additional data or training. We can then fine-tune the base model on a new domain and readapt it to instruction following with the reverse engineered adapter. RE-Adapt and our low-rank variant LoRE-Adapt both outperform other methods of fine-tuning, across multiple popular LLMs and datasets, even when the models are used in conjunction with retrieval-augmented generation.

## 1 Introduction

Large Language Models (LLMs) require a significant investment to develop and train, requiring resources available to only a limited number of organizations. For instance, Meta's Llama-3 family of models was trained using two custom-built compute clusters, each containing 24,000 high-end GPUs (Meta, 2024). Parameter Efficient Fine Tuning (PEFT) enables resource-efficient downstream customization of existing LLMs for new domains by adjusting a relatively small number of parameters while keeping the majority unchanged. However, an important distinction exists between the types of model used for further fine-tuning. It is common for LLM producers to release two versions of a model, one which is *pretrained* on a general task such as next-token prediction and an *instruct* version which is then continued trained on annotated data to learn how to follow instructions or respond to queries in a preferential manner (Touvron et al., 2023; Jiang et al., 2023; Almazrouei et al., 2023; Banks & Warkentin, 2024). The availability of both versions introduces a choice for organizations wanting to adapt a model to their custom task or domain.

While an instruction-tuned model is generally more capable for popular tasks, the majority of data available for additional fine-tuning is unlabeled, lacking the annotations expected from instruct models. This poses a significant problem as annotation by the downstream organization can be too difficult, expensive, or error-prone (Fredriksson et al., 2020; Desmond et al., 2021). Additional fine-tuning can also degrade the performance of the instruction-tuned model outside of the new fine-tuning distribution (Kotha et al., 2024). On the other hand, pretrained models can be easily fine-tuned with unlabeled text but lack the additional capabilities of their instruct counterparts.

To address this dilemma, we seek the ability to fine-tune existing LLMs on unlabeled text while retaining the capabilities from pre-existing instruction-tuning. We draw inspiration from *adapters*, sets of learnable parameters added to an existing model for fine-tuning (Bapna & Firat, 2019; Houlsby et al., 2019). We make the key observation that **the difference in weights between an instruction-tuned and corresponding pretrained model is effectively an adapter**. Isolating the information learned from instruction-tuning into this *Reverse Engineered (RE)-Adapter* enables fine-tuning of the pretrained model without directly impacting the instruction-tuning. The fine-tuned model can then be readapted with the instruction following capabilities (Figure 1).

We apply our RE-Adapt approach to fine-tune several popular LLM variants with unlabeled data from new domains. We use both closed- and open-book question-answering to measure their ability to incorporate the new domain-specific information without decreasing the previous capabilities out-of-domain. Specifically, we:

Figure 1: In RE-ADAPT, an *instruction adapter* is isolated by differencing weights between instruct (🎓) and pretrained (🎓) versions of a model and reapplied to the pretrained model after fine-tuning.

- Explore the differences in parameters between pretrained and instruct models and their use as instruction adapters;

- Quantify RE-ADAPT's effectiveness to leverage unstructured knowledge for question answering in new domains under both context-free and retrieval-augmented scenarios;

- Introduce *partial adaptation*, a technique for scaling the strength of adapters for fine-grain control of knowledge priorities; and

- Demonstrate that RE-Adapters are *effectively* low-rank, showing that low-rank RE-Adapters (LoRE-Adapters) are capable of similar performance using up to 5x fewer parameters.

## 2 BACKGROUND

### 2.1 ADAPTERS

Adapters (Bapna & Firat, 2019; Houlsby et al., 2019) have played an important role in the context of transfer learning for language models in recent years, particularly for fine-tuning pretrained models which are too large to fully train on commodity hardware. The concept introduced by Houlsby et al. (2019) provides a lightweight alternative to full fine-tuning through the augmentation of models with small modular sets of trainable parameters. Adapters have been useful for enabling the use of pretrained models on new tasks (Pfeiffer et al., 2021; Karimi Mahabadi et al., 2021; Rücklé et al., 2021), new domains (Malik et al., 2023; Schopf et al., 2023; Diao et al., 2023), and adapting to multiple languages (Chronopoulou et al., 2023b; Üstün et al., 2022; Parovic et al., 2023).

Low-Rank Adapters (LoRA) (Hu et al., 2022) are a particularly parameter efficient adaptation technique which adds a low-rank matrix to the weights of existing layers. Because the adapter is low-rank it can be represented as the product of two much smaller matrices, significantly lowering the number of trainable parameters. Weight-Decomposed Low-Rank Adaptation (DoRA) is an extension to LoRA with superior performance and similar efficiency (Liu et al., 2024). Liu et al. (2024) achieve this by decomposing the pretrained weights into both magnitude and direction components, applying LoRA for directional fine-tuning only. Important to this work, adapters learned with either LoRA or DoRA can be represented as a single matrix which captures the information learned during fine-tuning. The pretrained model is then adapted by simply adding the new matrix to the existing weights. We leverage DoRA to fine-tune our models on a new domain, and take inspiration from the additive nature of these techniques to derive our reverse engineered adapters.

Several approaches have been developed which utilize the mixing or combination of adapters to benefit from diverse tasks or domains Pfeiffer et al. (2021); Rücklé et al. (2021); Wang et al. (2022); Chronopoulou et al. (2023a); Fleshman et al. (2024); Zadouri et al. (2024) or for parameter efficient federated learning (Babakniya et al., 2023; Sun et al., 2024). One method to categorize these approaches is by the mechanism used for combining the adapters. Either a weighted combination of adapters is applied to the base model (Chronopoulou et al., 2023a; Fleshman et al., 2024; Babakniya et al., 2023; Sun et al., 2024) or another set of parameters are used to learn adapter interactions

(Pfeiffer et al., 2021; Rücklé et al., 2021; Wang et al., 2022; Zadouri et al., 2024). We focus on the former, as we reframe instruction-tuned models as the summation of a pretrained model with an instruction adapter. We add new knowledge by combining domain-specific and instruction adapters via linear combination. As highlighted by Sun et al. (2024), separate adapters can be incompatible when averaged. Chronopoulou et al. (2023a) and Fleshman et al. (2024) try to mitigate this by initializing adapters with the same random weights, and Sun et al. (2024) by doing the same through a data driven approach. Neither option is applicable here, as we have no control over the instruction adapter. This motivates our new approach for *partial adaptation* which we introduce in Section 3.

## 2.2 INSTRUCT MODELS

Some of the most capable LLMs are *instruct* variants, pretrained on massive amounts of unannotated text and further trained on curated datasets with a combination of instruction-tuning (Mishra et al., 2022; Wei et al., 2022; Ouyang et al., 2022b; Sanh et al., 2022) and Reinforcement Learning from Human Feedback (RLHF) (Christiano et al., 2017; Stiennon et al., 2020). For example, Llama-3 was pretrained on 15T tokens and the instruct version continued training with a combination of supervised fine tuning (SFT), rejection sampling, proximal policy optimization (PPO), and direct preference optimization (DPO) (Meta, 2024). Open-source LLM producers generally release both the instruct versions as well as the pretrained models from which they were derived (Jiang et al., 2023; Almazrouei et al., 2023; Banks & Warkentin, 2024; Meta, 2024). Access to the pretrained LLM allows users to customize the model to a new task or domain while taking advantage of the large investment required for pretraining. Fine-tuning the instruct model directly is generally avoided due to *catastrophic-forgetting*, a phenomenon where models lose previous abilities with subsequent rounds of continued training (McCloskey & Cohen, 1989; Kotha et al., 2024). This is unfortunate, as few organizations have the resources to conduct fine-tuning at the scale of the original instruction-tuned models. In this work, we explore methods of fine-tuning LLMs which take advantage of both the pretraining and instruction-tuning of existing LLMs. We specifically design our approach to minimize forgetting while fine-tuning instruction-capable models with unlabeled text.

## 2.3 MODEL ARITHMETIC

Previous works have looked at the ability to arithmetically manipulate models to isolate certain behaviors (Ilharco et al., 2023; Mitchell et al., 2024). Ilharco et al. (2023) constructed *task vectors* by differencing weights between a pretrained model and several corresponding models each fine-tuned for a particular task. They observed for their models that task vectors are almost orthogonal to each other, preventing interference and allowing combinations of the vectors for negating certain behaviors, improving multi-task performance, or performing well on new tasks via more complicated task analogies (Ilharco et al., 2023). We similarly solve for our reverse engineered adapter with a simple differencing, but using a single LLM fine-tuned for multi-task instruction-following. By effectively isolating instruction-tuning into an adapter, we allow further fine-tuning of pretrained models, maximizing knowledge acquisition before readapting their ability to follow instructions. We introduce an optional step for reducing the rank of our RE-Adapter, lowering memory requirements while maintaining or improving performance in some scenarios. Unlike task vectors, our multi-purpose RE-Adapters are not assumed to be orthogonal to new training domains. We introduce a technique for mitigating potential interference in Section 3 by controlling the adaptation strength.

Mitchell et al. (2024) developed an alternative approach for isolating pretraining knowledge from fine-tuned behaviors which they call *emulated fine-tuning*. Instead of differencing model weights, emulated fine-tuning considers the difference in outputs between pretrained and fine-tuned versions of a model. By combining this difference with the output of a larger pretrained model, Mitchell et al. (2024) found that they could benefit from the additional pretraining knowledge while still solving the task of the smaller model. Their technique could be extended to meet our goal but requires the storage and forward pass of multiple models for inference. Our approach isolates new knowledge and instruction-following into separate adapters, merged into a single model at no extra cost.

## 3 PARTIAL ADAPTATION

We detail our main methods in Section 4, but first we introduce a technique for controlling the strength of adaptation. Consider a model with weights $\mathbf{W}$ and an adapter $\mathbf{A}$ used to fine-tune the model on a new domain. Using additive adapters such as LoRA or DoRA, the combined weights:

$$\hat{\mathbf{W}} = \mathbf{W} + \mathbf{A} \tag{1}$$

are then used for inference (Hu et al., 2022; Liu et al., 2024). We make the observation that the resulting model assigns equal weight to the original parameters and the new adapter, which is generally trained with significantly less data than the original weights. This potentially leads to overfitting in the new domain and degradation of performance in the general setting. These issues compound in situations where multiple adapters are combined. Both Chronopoulou et al. (2023a) and Fleshman et al. (2024) discuss complications arising from mixing adapters, especially if they were not initialized with the same values to encourage compatibility.

To mitigate these challenges we propose a technique for *partial adaptation* which introduces a post-hoc scaling factor for each fine-tuned adapter. Importantly, Equation 1 is still used during fine-tuning, but inference becomes:

$$\hat{\mathbf{W}} = \mathbf{W} + \lambda\mathbf{A} \tag{2}$$

where $0 \leq \lambda \leq 1$ is used to scale the strength of adaptation. In our experiments, we find that partial adaptation improves performance when using either single or multiple combined adapters.

## 4 REVERSE ENGINEERED ADAPTATION

Here we describe Reverse Engineered Adaptation (RE-ADAPT), our approach to solve the challenge of updating an instruction-tuned model with unlabeled text without degrading the ability of the model to follow instructions. In Section 5, we demonstrate the effectiveness of this approach for closed-book and retrieval-augmented question answering over new domains.

### 4.1 RE-ADAPTERS

First consider two language models: $\mathbf{T_\Phi}$, which has been pretrained with parameters $\mathbf{\Phi}$; and $\mathbf{T_\Theta}$, having the same architecture as $\mathbf{T_\Phi}$ but with parameters $\mathbf{\Theta}$ updated from the pretrained parameters $\mathbf{\Phi}$ via instruction-tuning. Given these models, we can solve for the RE-Adapter parameters $\mathbf{\Delta}$ using:

$$\mathbf{\Delta} = \mathbf{\Theta} - \mathbf{\Phi} \tag{3}$$

to isolate the information learned during instruction-tuning. Next, we augment the pretrained model $\mathbf{T_\Phi}$ with a learnable adapter $\mathbf{\Psi}$ and fit $\mathbf{T_{\Phi+\Psi}}$ on a new domain by only updating the adapter weights $\mathbf{\Psi}$. We refer to $\mathbf{\Psi}$ as the *knowledge adapter*. We utilize DoRA to fit $\mathbf{\Psi}$ in our experiments, but any fine-tuning approach is applicable. We construct our final model $\mathbf{T_\Omega}$ with parameters:

$$\mathbf{\Omega} = \mathbf{\Phi} + \alpha\mathbf{\Psi} + \beta\mathbf{\Delta} \tag{4}$$

where $\alpha$ and $\beta$ are the scaling factors for the partial adaptation of $\mathbf{\Psi}$ and $\mathbf{\Delta}$ respectively. We find that scaling down the strength of the knowledge adapter $\mathbf{\Psi}$ and RE-Adapter $\mathbf{\Delta}$ with partial adaptation leads to better performance in instruction-based tasks related to the new domain while maintaining or slightly improving on the performance of the original instruction-tuned model out-of-domain. Except for measuring their sensitivity, we simply set $\alpha$ and $\beta$ to an equal weighting of 0.5 in our experiments.

### 4.2 LoRE-ADAPTERS

Inspired by LoRA, we explore the intrinsic dimensionality of RE-Adapters and their ability to be represented by low-rank approximations. The Eckart-Young-Mirsky theorem establishes the truncated singular value decomposition (SVD) as the best low-rank approximation of matrices under the Frobenius norm (Eckart & Young, 1936). We compute the SVD of the RE-Adapter $\mathbf{\Delta}$ from Equation 3 which yields $\mathbf{\Delta} = \mathbf{USV}^\intercal$ with the diagonal of $\mathbf{S}$ containing the singular values of $\mathbf{\Delta}$ sorted by magnitude, and with $\mathbf{U}$ and $\mathbf{V}$ the corresponding left and right singular vectors. We then

compute the percentage of variance explained by each dimension by squaring the singular values and re-normalizing the results to sum to 1. The cumulative explained variance $v$ at rank $k$ is then:

$$v_k = \Sigma_{i=0}^{k} \frac{\sigma_i^2}{\Sigma \sigma_j^2} \tag{5}$$

where $\sigma_i$ is the $i$th singular value. We replicate this analysis for multiple modern LLMs and find that the majority of total variation in parameters can be represented at low-rank. For example, Figure 2 displays the cumulative explained variance plots for three layers from the RE-Adapter derived from Llama-3, where we see more than half of the variance in these layers can be captured by a rank 128 approximation. This suggests the potential for a low-rank RE-Adapter (LoRE-Adapter).

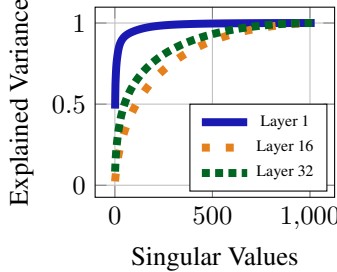

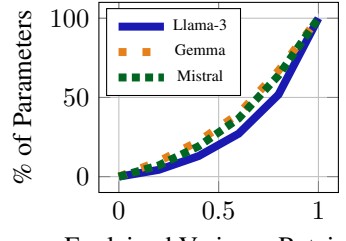

Figure 2: Cumulative explained variance for singular values from Llama-3 RE-ADAPT *k_proj* layers.

Figure 3: Percent of parameters used for LORE-ADAPT with varying threshold of explained variance.

We can convert a RE-Adapter into a LoRE-Adapter by representing each layer with its truncated SVD (Sharma et al., 2024). Specifically, we truncate each layer to the rank that captures a total explained variance above a user-defined threshold $\tau$. Figure 3 shows the relationship between $\tau$ and the reduction in total parameters when using Llama-3 models to derive the adapter. As $\tau$ increases we maintain a higher percentage of the original parameters. We use LoRE-Adapters with $\tau = 0.5$ for the experiments in this work and see similar or better performance when compared to RE-ADAPT while using up to 5x less parameters. Like LoRA, the savings in memory is beneficial in cases where several LoRE-Adapters are swapped in and out of the same model.

## 5 EXPERIMENTS

We quantify the effectiveness of RE-ADAPT using question-answering (QA), a task suitable for measuring knowledge acquisition from unlabeled data for which instruction-tuned models perform better than their pretrained counterparts due to significant portions of instruction-following datasets being QA-based (Ouyang et al., 2022a; Wang et al., 2023; Taori et al., 2023). Specifically, we want to see if RE-ADAPT is better than alternatives for adding knowledge from data not annotated with question-answer pairs. We would like the resulting model to do well answering questions about the new domains, while maintaining the level of performance of the original instruction-tuned model when answering unrelated questions.

### 5.1 MODELS

We replicate all experiments using the pretrained and instruct versions from the Gemma-7B (Banks & Warkentin, 2024), Llama-3-8B (Meta, 2024), and Mistral-7B (Jiang et al., 2023) family of LLMs using the HuggingFace API (Wolf et al., 2020). We utilize the parameter efficient fine-tuning library (Mangrulkar et al., 2022) for adding DoRA (Liu et al., 2024) knowledge adapters to each of these models. We perform all fine-tuning and inference with a single 80GB A100 GPU. We include hyper-parameters and other details of our fine-tuning in Appendix A.

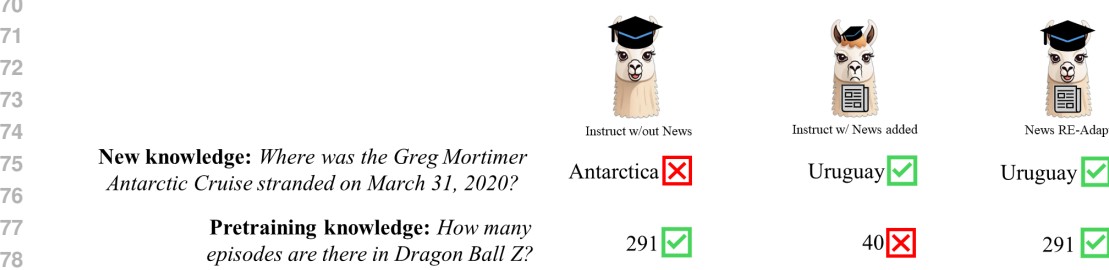

**New knowledge:** *Where was the Greg Mortimer Antarctic Cruise stranded on March 31, 2020?*

**Pretraining knowledge:** *How many episodes are there in Dragon Ball Z?*

Figure 4: RE-ADAPT enables the addition of new knowledge to an instruction-tuned model, without degrading capabilities on knowledge from pretraining.

In Section 5 we compare RE-ADAPT and LoRE-ADAPT with the pretrained and instruct models of each family, as well as pretrained and instruct models fine-tuned with DoRA on the new domains. We perform experiments for closed-book QA as well as QA with retrieval-augmented generation (RAG).

## 5.2 DATA

Kotha et al. (2024) showed that fine-tuning degrades performance outside of the fine-tuning distribution. We hypothesize that our approach mitigates this issue by isolating existing instruction-tuning from additional fine-tuning. We test this by measuring the changes in question-answering performance when various fine-tuning strategies are used to update models with unlabeled data. An optimal approach would benefit from the new knowledge when asked related questions, without losing the ability to answer unrelated questions.

We explore this hypothesis by fine-tuning models in two different settings. We use English WMT News Crawl (Kocmi et al., 2022) articles published in the year 2020 as our first fine-tuning distribution.[1] These articles provide non-annotated information which we capture through DoRA adapters trained for next-token-prediction. We evaluate how well this knowledge is acquired by using the resulting models to answer related questions from the StreamingQA dataset (Liška et al., 2022), which contains 21,681 QA pairs derived from our subset of articles.[2]

We use the evidence passages from RetrievalQA (Zhang et al., 2024) as our second fine-tuning distribution and measure performance on the corresponding questions from the same dataset.[3] Zhang et al. (2024) curated the dataset by compiling the subset of questions from five other QA benchmarks for which GPT-4 (OpenAI et al., 2024) is unable to answer without access to external knowledge. The questions were selected with the goal of having the corresponding knowledge absent from current LLMs, making this dataset especially challenging in the closed-book setting.

To measure any performance degradation from fine-tuning, we also evaluate our models using a short-answer subset of the Natural Questions dataset (Kwiatkowski et al., 2019) which is unrelated to either fine-tuning distribution.[4] We use these questions as a control to measure performance before and after fine-tuning our models on the other domains. We would like our approach to result in improved performance when answering questions related to the fine-tuning data without a reduction in performance on the unrelated Natural Questions Figure 4.

## 5.3 EVALUATION

We observe that instruction-tuned models will generally answer questions in long form, often repeating the question and providing additional helpful context. An example of this behavior is shown in Table 1 where the model is asked for the number of episodes in a popular television series. Here we see the reference answer is 291, which Llama-3 gets correct, but with a response containing full sentences and additional information to clarify its position.

---

[1] Available at `https://data.statmt.org/news-crawl/README` under CC0 license.

[2] Available at `https://github.com/google-deepmind/streamingqa` under CC-BY 4.0 license.

[3] Available at `https://huggingface.co/datasets/zihanz/RetrievalQA` under MIT license.

[4] Available at `https://huggingface.co/datasets/natural_questions` under CC-BY-SA 3.0 license.

Table 1: Example from Natural Questions with a truncated response. Llama-3's full response includes more details per country.

| Question | how many episodes are there in dragon ball z? |
|---|---|
| Answer | 291 |
| Llama-3 | There are a total of 291 episodes in the original Japanese version of Dragon Ball Z. However, the episode count can vary depending on the version and the country. |

Popular QA metrics such as Rouge-L (Lin, 2004) or exact match would penalize Llama-3 for not being precise. To alleviate this concern we evaluate using Rouge-L's recall, which is the percentage of the longest common sub-sequence of the reference answer found in the model's response. We additionally measure a version of exact match which looks for the exact reference answer anywhere in the response. In both cases, if the reference answer is in the response the score will be 1. If the answer is partially correct then exact match will be 0, but Rouge-L will provide partial credit.

### 5.4 CLOSED-BOOK QA

In our first experiment, we conduct QA evaluation in a closed-book setting where the models must provide an answer given nothing but the question. We explore how RE-ADAPT behaves in this setting with varying partial adaptation scaling factors. Figure 5 shows the QA performance of Llama-3 using a fixed factor of 1.0 for the knowledge adapter with varying scaling factors for the RE-Adapter. We find that partial adaptation with a factor of 0.5 for both the knowledge adapter and instruction adapter provides

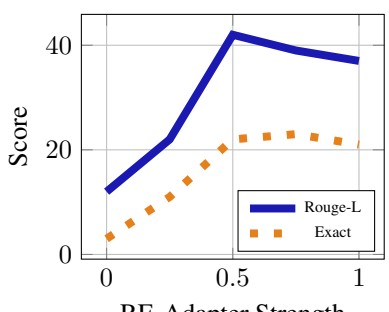

Figure 5: StreamingQA performance as RE-Adapter is added to fine-tuned Llama-3 model with varying strengths.

robust results across models and datasets when using both RE-ADAPT and LORE-ADAPT. We use an explained variance threshold $\tau = 0.5$ for our LoRE-Adapters. The resulting percentage of original parameters for each model are: Llama-3 (19.2%), Gemma (30.2%), and Mistral (27.1%).

The closed-book performance of all models across datasets is shown in Table 2. Both RE-ADAPT and LORE-ADAPT outperform the pretrained and instruction-tuned models on StreamingQA and RetrievalQA, even when those models are fine-tuned on the corresponding News Crawl or RetrievalQA passages. As expected, the pretrained models perform worse, although fine-tuning on the unlabeled data does improve the QA ability of both pretrained and instruct models in the domain used for adaptation. These in-domain results suggest that our approach is superior for knowledge acquisition. Next, we will discuss the impact fine-tuning has on general QA performance by looking at results on the out-of-domain Natural Questions dataset used as a control.

The closed-book results for the Natural Questions dataset on the right side of Table 2 demonstrate the issues with fine-tuning instruct models with non-annotated data, resulting in models that perform worse in their original setting. While fine-tuning on News Crawl or Retrieval QA passages improved the instruct models on the corresponding QA datasets, the majority of models saw a decrease in performance on Natural Questions. RE-ADAPT alleviates this problem by using the data from the new domain to only fine-tune the pretrained model, keeping the instruction-tuning intact. Using our approach, the resulting models performed significantly better on the fine-tuning distribution without a performance degradation out-of-domain. In fact, **RE-ADAPT and LORE-ADAPT performed better than the original instruction-tuned models out-of-domain**. This improvement indicates that instruction-tuning likely degrades knowledge from pretraining; an issue our approach mitigates through partial adaptation. We confirm this suspicion by applying RE-ADAPT to Llama-3 without any additional fine-tuning. This allows us to produce instruct models with instruction-tuning strengths ranging from 0 (the pretrained model) to 1 (the instruct model).

Table 2: Closed-book QA performance. The QA dataset being evaluated is listed above the dataset used for fine-tuning DoRA adapters. R-L indicates Rouge-L and EM indicates exact match.

| | Model | StreamingQA News Crawl | | RetrievalQA RQA Passages | | Natural Questions News Crawl | | RQA Passages | |
|---|---|---|---|---|---|---|---|---|---|
| | | R-L | EM | R-L | EM | R-L | EM | R-L | EM |
| Llama-3 | Pretrained | 9 | 0 | 1 | 0 | 10 | 3 | 10 | 3 |
| | Pretrained + DoRA | 12 | 3 | 3 | 2 | 10 | 4 | 14 | 7 |
| | Instruct | 33 | 19 | 5 | 3 | 46 | 34 | 46 | 34 |
| | Instruct + DoRA | 38 | 22 | 7 | 4 | 39 | 22 | 37 | 27 |
| | LoRE-Adapt (Ours) | 46 | 26 | **10** | **6** | 51 | 34 | 53 | 35 |
| | RE-Adapt (Ours) | **46** | **27** | 9 | 6 | **52** | 34 | **54** | **36** |
| Gemma | Pretrained | 11 | 2 | 1 | 0 | 10 | 3 | 10 | 3 |
| | Pretrained + DoRA | 19 | 4 | 1 | 0 | 7 | 1 | 10 | 2 |
| | Instruct | 20 | 9 | 2 | 1 | 26 | 12 | 26 | 12 |
| | Instruct + DoRA | 31 | 18 | 5 | 3 | 26 | 12 | 28 | 14 |
| | LoRE-Adapt (Ours) | 31 | 15 | **7** | **4** | 24 | 14 | **30** | **20** |
| | RE-Adapt (Ours) | **33** | **18** | 6 | 4 | **26** | **17** | 28 | 17 |
| Mistral | Pretrained | 17 | 5 | 2 | 0 | 14 | 5 | 14 | 5 |
| | Pretrained + DoRA | 22 | 8 | 2 | 1 | 14 | 5 | 15 | 6 |
| | Instruct | 29 | 16 | 4 | 2 | 33 | 22 | 33 | 22 |
| | Instruct + DoRA | 36 | 21 | 6 | 5 | 27 | 13 | 33 | 18 |
| | LoRE-Adapt (Ours) | **39** | **24** | **7** | **5** | **39** | **24** | **42** | **28** |
| | RE-Adapt (Ours) | 37 | 22 | 6 | 4 | 37 | 23 | 41 | 27 |

We find that **we can improve existing instruct models with zero additional training by simply scaling down the strength of instruction-tuning** Figure 6. Combined, these results demonstrate the effectiveness of RE-ADAPT for knowledge acquisition with minimal *forgetting*.

## 5.5  RE-ADAPT WITH RAG

Retrieval-augmented generation (RAG) Lewis et al. (2020) is a popular alternative for utilizing new data with instruction-tuned models. Instead of altering the model directly, RAG maintains a database of all text and retrieves relevant documents to include in the prompt as context. This begs the question, is RE-ADAPT still beneficial if the new data is already available via RAG?

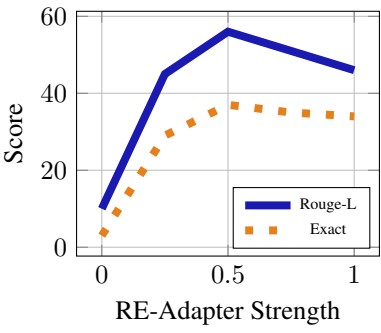

Figure 6: Natural Questions performance as the RE-Adapter is added to pretrained Llama-3 with varying strengths.

To answer this question, we replicate our experiments on StreamingQA and RetrievalQA, using a BM-25 index (Robertson & Zaragoza, 2009) to retrieve the most relevant passage to be used as context for the models. In practice, RAG setups can retrieve more than one document, but each question in our datasets can be answered from a single passage, and therefore we avoid known issues which RAG can face when too much context is provided to the models (Liu et al., 2023; Barnett et al., 2024; Gao et al., 2024). A poor retriever could bias results in our favor, so we also repeat the experiment using an oracle retriever. Instead of performing a heuristic search, the oracle retriever directly selects the passages capable of answering the question as context. While this idealized retriever is unrealistic in practice, it allows us to further isolate the benefit of combining RAG with fine-tuning by eliminating any impact from imperfect retrieval.

The RAG results are shown in Table 3. Again we see significant improvements when using RE-ADAPT and LoRE-ADAPT even in this RAG setting where the model should already have access

to the relevant information needed to answer the questions. The BM-25 search retrieved the correct document with approximately 73% accuracy across models. Using RE-ADAPT to incorporate the data outside of RAG alleviates the shortcomings of the retriever. However, RE-ADAPT also improved results when using the oracle, suggesting that adding domain knowledge with an adapter also reduces incorrect interpretations of the context retrieved via RAG.

Table 3: QA performance when using RAG with BM25 and (Oracle) retrievers.

| | Model | StreamingQA | | RetrievalQA | |
|---|---|---|---|---|---|
| | | Rouge-L | Exact Match | Rouge-L | Exact Match |
| Llama-3 | Pretrained | 38 (59) | 27 (48) | 13 (16) | 11 (14) |
| | Instruct | 55 (57) | 54 (58) | 14 (30) | 16 (32) |
| | LoRE-ADAPT (Ours) | **69 (74)** | 58 (64) | **24 (37)** | **21 (31)** |
| | RE-ADAPT(Ours) | 68 (71) | **59 (64)** | 19 (36) | 18 (30) |
| Gemma | Pretrained | 39 (41) | 28 (29) | 4 (26) | 3 (23) |
| | Instruct | **52 (56)** | 48 (53) | 17 (24) | 16 (24) |
| | LoRE-ADAPT (Ours) | 46 (50) | 49 (55) | 12 (17) | 18 (27) |
| | RE-ADAPT (Ours) | 50 (55) | **50 (56)** | **21 (30)** | **18 (28)** |
| Mistral | Pretrained | 33 (38) | 26 (30) | 18 (12) | 16 (10) |
| | Instruct | 49 (52) | 50 (56) | 14 (23) | 19 (28) |
| | LoRE-ADAPT (Ours) | 54 (58) | **55 (61)** | **18 (23)** | 20 (28) |
| | RE-ADAPT (Ours) | **55 (58)** | 55 (60) | 15 (24) | **20 (29)** |

## 6 DISCUSSION

Combined, our results demonstrate RE-ADAPT's effectiveness at incorporating new knowledge into existing LLMs without having to discard previous instruction-tuning. Our methods increase QA performance by a greater amount when compared to traditional fine-tuning strategies. We also find that our approach improves RAG based systems, even in the most optimistic case of perfect retrieval. Our improved results outside of the fine-tuning distribution suggest that we can recover additional pretraining knowledge by reducing the strength of instruction-tuning through partial adaptation. Importantly, an improvement is seen without any additional fine-tuning of the underlying models. These results encourage additional future research into controlling the competing priorities of knowledge acquisition and general problem solving capability.

## 7 CONCLUSION

In this work, we presented RE-ADAPT, a new approach for adding knowledge to existing instruction-tuned models. RE-ADAPT isolates the differences between an instruction-tuned model and its pretrained counterpart in order to preserve instruction-following capabilities during additional fine-tuning on unlabeled data. We demonstrated that our approach outperforms fine-tuning pretrained or instruction-tuned models directly, which otherwise causes performance to degrade outside of the new fine-tuning domain. Our findings are robust across three state of the art large language models.

We achieved our best performance using *partial adaptation*, a new method for controlling the strength of adaptation at inference time when using single or combined adapters. We found that partially adapting instruction-tuned models improved QA performance without any additional fine-tuning.

We also analyzed the spectrum of RE-ADAPT's weight matrices, constructing a low-rank variant of our approach, LoRE-ADAPT, which captures the majority of variation in the instruction-tuning weights at a much lower rank. LoRE-ADAPT performed similarly to RE-ADAPT with occasional out-performance, while decreasing the number of parameters by as much as 5x in our experiments.

Finally, we demonstrated that RE-ADAPT improves performance even when the information required to answer questions is available via retrieval augmented generation. Combined, our results suggest RE-ADAPT is an effective approach for infusing new knowledge into instruction-tuned LLMs.

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

## A  FINE-TUNING DETAILS

We include the settings for training our DoRA adapters in Table 4. All adapters were trained on a single NVIDIA A100 GPU with 80GB of memory.

Table 4: Training details.

| Setting | Value |
|---|---|
| LoRA Layers | all-linear |
| LoRA Rank | 64 |
| LoRA Alpha | 128 |
| LoRA Dropout | 0.05 |
| DoRA | True |
| Batch Size | 20 |
| Epochs News Crawl | 10 |
| Epochs RetrievalQA | 3 |
| Optimizer | AdamW |
| Learning Rate | 0.0002 |
| Schedule | Linear |

## B  PROMPTS USED

Each LLM can use unique prompting roles and tokens when constructing prompts. We utilize the huggingface *tokenizers* library to ensure our prompts follow the correct template.

The Llama-3 instruct models use a combination of system, user, and assistant roles while Gemma and Mistral only use user and assistant. Our prompts where constructed using the following formats:

**Llama-3 Closed-Book QA**
system: *Answer the following question.*
user: *<question>?*

**Llama-3 RAG**
system: *Answer the following question given this context: <context>.*
user: *<question>?*

**Gemma and Mistral Closed-Book QA**
user: *<question>?*

**Gemma and Mistral RAG**
user: *Answer the following question given this context: <context>\nQuestion: <question>?*

