# OpenReview forum: "RE-Adapt: Reverse Engineered Adaptation of Large Language Models"
_ICLR.cc/2025/Conference — ICLR 2025 Conference Withdrawn Submission_

### Official Review · Reviewer_qowB · 2024-10-28

**Soundness:** 2
**Presentation:** 3
**Contribution:** 2
**Rating:** 3
**Confidence:** 5

**Summary:**

The paper investigates strategies for continual adaptation of instruction-tuned LLMs to new domains and proposes RE-Adapt, reverse-engineered adaptation of LLMs. The approach takes inspiration in weight arithmetic and produces a domain-adapted instructed model as a linear combination of a pretrained LLM, a Reverse-engineered adapter (difference between the weights of an instructed and the pretrained LLM), and a Knowledge adapter (weight change resulting from tuning a pretrained LLM on a new domain). The paper also suggests a partial adaptation strategy, i.e. tuning a scalar multiplicative weight for each term in the weight arithmetic, and LoRE-Adapt, a modification of Re-Adapt which uses a low-rank approximation of the Reverse-engineered adapter obtained through singular value decomposition. The proposed method is validated on two question answering datasets, while monitoring the performance on Natural Questions as a generic dataset.

**Strengths:**

* A highly-relevant research direction of adapting LLMs to new domains
* An interesting idea of constructing domain-adapted instructed models by task arithmetic. The simplicity of the method makes it easily applicable in practice to any domain of interest.
* The paper is well-written and easy to follow
* Well-structured and comprehensive Related Work
* The method is validated for three different LLMs

**Weaknesses:**

1. Insufficient empirical validation of the proposed approach.
     - The paper conducts experiments on a single task (question answering), with two adaptation datasets (StreamingQA and RetrievalQA) and one general QA dataset (Natural Questions). Since the main goal of the proposed approach is to build a model capable of following instructions, a validation of the final model on a broader set of tasks is required, e.g. including open-ended instruction following, summarization, or multiple-choice QA. This concerns both instruction following capabilities on new domains and initial model capabilities, i.e. validation solely on NQ is not enough to demonstrate the preservation of initial instructed LLM capabilities. It is possible to create multi-task evaluation beds e.g. for a biomedical domain with Pubmed as domain data [1-4].
     -  I am not sure if RetrievalQA is a good candidate domain for the paper’s experiments, since performance on this dataset without RAG is very low (and improvements by the proposed method  are also small, i.e. 1-2 % exact match), and because evidence passages from RetrievalQA do not necessarily form a well-defined domain.
    - Tables 2 and 3 miss a baseline $\Phi + 0.5 \Delta$, to disambiguate the influence of downscaling the Re-adapter (shown to be effective in Figure 6) and of learning the domain knowledge in the domain adapter. For example, in Table 3 the domain knowledge comes from using RAG, so this baseline can potentially perform as good as Re-Adapt / LoRE-Adapt.
    - It would be also an advantage if the proposed approach was validated for more than one adaptation approach, e.g. simple LoRA, as it is widely used in practice.
2. The partial adaptation approach is not novel as the same technique of applying a scalar factor, tuned on the validation set, was used e.g. in (Inarco, 2023).
3. The experiments are conducted with a fixed learning rate, which weakens a simple baseline of finetuning an instructed LLM on the domain data. As shown in [5, 6], simply reducing an LR may help to avoid catastrophic forgetting when tuning LLMs.  According to Table 4 in Appendix, the paper uses an LR of 2e-4, which is too high to preserve initial model capabilities according to [6].
4. Deeper details on the experimental setup could be provided, such as the used decoding strategy at inference, metric details in case some preprocessing was used (e.g. lower casing candidate and ground truth answer), preprocessing details for the datastore in RAG (e.g. how passages for retrieval are constructed), or which parameters are used in bm25.

[1] Anastasios Nentidis et al. Overview of BioASQ 2023: The eleventh BioASQ challenge on Large-Scale Biomedical Semantic Indexing and Question Answering. CLEF 2023. (Factoid / list / yes-no questions and summarization on the biomedical domain.)

[2] Xinlu Zhang et al. AlpaCare: Instruction-tuned Large Language Models for Medical Application. arxiv 2023. (This paper collects a MedInstruct-test dataset which can be used for evaluating open-ended generation on the biomedical domain.)

[3] Qiao Jin et al. PubMedQA: A Dataset for Biomedical Research Question Answering. EMNLP 2019

[4] Hongjian Zhou et al. A Survey of Large Language Models in Medicine: Progress, Application, and Challenge. arxiv 2023

[5] Nadezhda Chirkova et al. Key ingredients for effective zero-shot cross-lingual knowledge transfer in generative tasks. NAACL 2024

[6] Nadezhda Chirkova et al.  Zero-shot cross-lingual transfer in instruction tuning of large language models. INLG 2024

**Questions:**

1. How long does SVD decomposition procedure take, compared to the DoRA tuning process (execution time)?
2. Line 244: “we truncate each layer to the rank that captures a total explained variance above a user-defined threshold $\tau$”. Did I understand correctly that the DoRA rank will differ between layers?

Additional comments

Experiments:

* Line 374: “RE-ADAPT and LORE-ADAPT performed better than the original instruction-tuned models out-of-domain. This improvement indicates that instruction-tuning likely degrades knowledge from pretraining”. The second sentence does not necessarily follow from the first one and would require additional experimental validation. For example, this improvement may simply come from a regularization effect of weight averaging [7]. The same regularization effect could be the reason for the improvement in Figure 6.
* Line 434: “RE-ADAPT also improved results when using the oracle, suggesting that adding domain knowledge with an adapter also reduces incorrect interpretations of the context retrieved via RAG.” Again the second part of the sentence does not necessarily follow from the first part and would require additional experimental validation. For example, the improvement may potentially come from scaling down the contribution of Re-adapter, same effect as in Figure 6.

[7] Pavel Izmailov et al. Averaging Weights Leads to Wider Optima and Better Generalization. UAI 2018

Text:

* Table 2: there are several cases when the baseline has the same performance as the proposed method, but is not highlighted bold. Example: Gemma / Instruct + DoRA / second column -> 18 should also be bold; there are several other cases.
* Unclear claim in line 37: “lacking the annotations expected from instruct models. ”
* Line 92: “Because the adapter is low-rank it can be represented as the product of two much smaller matrices, significantly lowering the number of trainable parameters.”: I think the premise and the conclusion should be reversed in this sentence, e.g. adapter is low-rank because it can be represented …
* Line 103: add \citet to put brackets around citations
* Line 261 and further in the text: “when answering unrelated questions.”: unclear term “unrelated questions”
* Line 284: remove “In section 5” as this is inside section 5.
* Line 300: “derived from our subset of articles”: what is “our subset”?
* Lines 295-300: I did not understand the connection between the StreamingQA dataset and WMT news crawl.
* Figure 5: I would suggest including similar plots for a fixed factor of 0.5 for a knowledge adapter, or varying a scalar factor for the knowledge adapter with the fixed scalar factor for Re-adapter.

---

### Official Review · Reviewer_VgZB · 2024-11-02

**Soundness:** 1
**Presentation:** 3
**Contribution:** 2
**Rating:** 3
**Confidence:** 4

**Summary:**

This paper proposes RE-Adapt and its variant LoRE-Adapt as methods for enhancing downstream tasks in specific domains by injecting domain knowledge into instruction models. Firstly, the authors extract the knowledge injected into the model during the SFT and RLHF stages by recording the parameter changes
of the instruction model compared to the pretrain model, and use this as an adapter. They then fine-tune a new adapter on the downstream tasks based on the pretrain model, and merge the two adapters. The authors have validated the effectiveness of their proposed method on several downstream tasks.

**Strengths:**

- The proposed method is straightforward, simple, and clear, making it easy for readers to understand. Additionally, the authors’ writing is fluent and well-executed.
- The authors have identified a problem that, while not highly focused on, is still of practical significance: the mismatch between instruction models and domain-specific downstream data.

**Weaknesses:**

- Lack of Novelty: The methods proposed by the authors, namely RE Adapt and LoRE-Adapt, are based on well-established concepts such as adding independent adapters to the model, utilizing Singular Value De composition (SVD) to extract key information from matrices, and com bining adapters through weighted averaging. These methods were widely accepted and recognized by the academic community as early as 2021 with the introduction of LoRA. In the paper, the authors do not present any novel insights or groundbreaking approaches to solving the problem, which makes the paper seem more like an exploration of existing methods on certain datasets rather than a demonstration of a new way to address challenges.
- Lacking Comparison Experiments: The experimental section lacks critical comparisons. The issue raised by the authors is the format mis match between downstream task data and the instruction model’s for mat. In fact, the authors could have easily constructed versions of exist ing datasets that match the instruction format through a text pipeline, and then fine-tuned the instruction model directly through various meth ods. This approach is more straightforward compared to the proposed method. The authors did not consider comparing their method with the restructured datasets, which I believe fails to demonstrate the practical
significance of their method.
- Lacking Discussion on Hyperparameters: As an important variant of the proposed method, LoRE-Adapt lacks experimental characterization of the impact of choosing different downsampling ratios 𝜏 on model per formance and inference cost.
- The paper lacks theoretical insights. While the experiments demonstrate a significant performance improvement with RE-ADAPT, there is insufficient theoretical support for the reasons behind this enhancement.
- Equation (4) introduces α and β as scaling factors for adjusting the strengths of the respective adapters, but there is a lack of experimental analysis discussing the balance between the knowledge adapter Ψ and the RE-Adapter Δ.
- LORE-ADAPT is a very interesting component, and many of its properties are quite intriguing. However, in terms of addressing the problem, both the principles and experimental results seem to be not that different from the RE-ADAPT method.
﻿
﻿Others
- The authors’ estimation of model parameters seems to be incorrect. In Figure 3, when 𝜏 = 1, the parameter size reaches 100%. In fact, due to the structural characteristics of LoRA, when 𝜏 = 1, the parameter size should be 200%, i.e., retaining two matrices of the same size as the original matrix, representing the A and B matrices in LoRA.
- From the experiments, LORE-ADAPT, the low-rank variant of RE-ADAPT, shows similar performance while reducing parameters by 4 to 5 times, and even outperforms RE-ADAPT in certain models (e.g., Mistral-7B). However, the paper lacks explanation and discussion on this observation. It would be beneficial for the authors to provide more details regarding LORE-ADAPT.

**Questions:**

- The method proposed by the authors primarily retains instruction-following abilities while preserving fine-tuning knowledge, so I would like to see how much impact this method has on instruction-following capabilities.
- The process of singular value decomposition (SVD) for delta does not clarify which parameter matrices were decomposed, making Figure 2 difficult to understand. Is it the entire parameter matrix?
- Why does an explained variance threshold of τ = 0.5 yield the best performance? The authors do not provide a reasonable explanation. Increasing the threshold leads to an increase in the number of parameters, but why does performance also decrease?
- The significant performance improvements brought by the proposed LoRE-ADAPT and RE-ADAPT are somewhat surprising, especially considering the additive process in Equation 4. Using the pretrained model + DoRA to train an adapter and then adding delta weights should not yield a performance significantly better than the pretrained model + DoRA alone. I hope the authors can provide more experimental results to demonstrate the effectiveness of LoRE-ADAPT and RE-ADAPT.

---

### Official Review · Reviewer_zsZy · 2024-11-03

**Soundness:** 2
**Presentation:** 3
**Contribution:** 3
**Rating:** 5
**Confidence:** 4

**Summary:**

This paper introduces a reverse-engineering adapter that enables adapting instruction-finetuned large language models (LLMs) to new domains while preserving their instruction-following capabilities. The authors model an instruction-tuned LLM as a combination of a pretrained model and an instruction adapter, effectively viewing the adapter as the weight difference between the pretrained and instruction-tuned models. A post-hoc scaling factor $𝜆$ is introduced to control the strength of the domain adaptation, with two variations presented—one with a low-rank adapter and one without. Experimental results on 7B models from three distinct LLM families, evaluated on closed and open-book question-answering tasks, show that the proposed approach outperforms traditional fine-tuning and retrieval-augmented generation (RAG) methods.

**Strengths:**

-  The paper addresses the important challenge of adding new knowledges to LLMs while preserving their instruction-following capabilities.
-  The method is well-grounded and aligns with existing literature on LLM arithmetic, leveraging the modular nature of model weights for adaptation.

**Weaknesses:**

- The experiments focus exclusively on question-answering (QA) tasks and do not assess whether the instruction-following capabilities of instruction-tuned models are maintained post-adaptation. Additionaly experiments on common benchmarks for instruction-tuned models are expected.
- A comparative analysis between the original instruction adapter obtained via reversed engineering and the instruction adapter after further domain-specific pretraining and instruction fine-tuning could help validate the authors’ hypothesis about the transferability of the instruction adapter.
- There is insufficient analysis of the chosen hyperparameters, particularly the sensitivity of the scaling factor $λ$, which could impact the generalizability of the approach.
- Experiments on different model sizes (<7B and >7B) would provide a more comprehensive understanding of whether the proposed method’s benefits hold at various scales.
- Tables 2 and 3 should report results to two decimal places. Currently, certain models appear to have identical performance (e.g., Gemma Instruct, Gemma Instruct + DORA, and Gemma Re-Adapt on the Natural Questions dataset all report a score of 26).

**Questions:**

1. How are the instruction-tuned models obtained? Are they officially released instruction-tuned models from each LLM family, or are they fine-tuned on downstream tasks? If the models were officially released, a more rigorous evaluation on established benchmarks might be needed to confirm if the instruction-following performance indeed degrades after the adaptation.
2. In Section 4.1, line 207, the authors mention conducting experiments on the sensitivity of scaling factors. However, no results for these experiments are provided in the results section.

---

### Official Review · Reviewer_DDd3 · 2024-11-05

**Soundness:** 2
**Presentation:** 3
**Contribution:** 2
**Rating:** 5
**Confidence:** 3

**Summary:**

This paper presents a novel fine-tuning method for large language models (LLMs) aimed at addressing the issue of catastrophic forgetting during adaptation to new domains. The proposed method, named RE-Adapt, integrates an instruction-following adapter with a domain-specific adapter. The instruction-following adapter is derived by comparing the foundation LLM with its instruction-tuned variant, while the domain-specific adapter is fine-tuned using DoRA. These adapters are combined during inference through weighted summation. Additionally, a variant called LoRE-Adapt substitutes the adapter with LoRA.

Experimental results on the StreamingQA and RetrievalQA benchmarks indicate that RE-Adapt outperforms baseline methods in in-domain adaptation performance. Furthermore, the proposed methods effectively mitigate the catastrophic forgetting problem compared to traditional fine-tuning pipelines (w/ DoRA in this paper), as demonstrated by out-of-domain results on the Natural Questions benchmark.

**Strengths:**

1. The paper is well-motivated, highlighting the challenge of catastrophic forgetting when adapting current LLMs to new domains or distributions. The RE-Adapt method provides a practical solution to this issue.
2. The LoRE-Adapt method, based on LoRA, demonstrates lower memory usage compared to RE-Adapt, which is significant for training efficiency, building on the inherent advantages of low-rank approximation.
3. The proposed methods are compatible with retrieval-augmented generation, showcasing their potential for integration with training-free approaches.

**Weaknesses:**

1. The applicability of the proposed method may be limited, as it relies on an instruction-tuned version of the original foundation model. While many open-source LLMs offer both versions, it remains unclear how RE-Adapt would perform if only the foundation model is available and we attempt to tune a weaker instruction-following model.
2. On line 201, the authors state that “any fine-tuning approach is applicable”, which is misleading. Directly employing a full-parameter fine-tuning method would prevent the combination of the tuned model's weights ($\Psi$) with the original weights ($\phi$) as described in Equation 4.
3. There is a lack of a LoRA baseline. Since LoRE-Adapt is a variant based on LoRA, it is essential to include a baseline that fine-tunes a pre-trained or instruction-tuned model using LoRA, in addition to the +DoRA baseline.
4.  The RAG setups in RetrievalQA should pose more significant challenges. Currently, the BM25 retriever searches only within in-domain data, while a more realistic scenario would involve a retrieval datastore created from diverse external data.
5. The proposed method combines different adapters and weights in a linear fashion. A previous study [1] systematically investigates the arithmetic operations of parameter-efficient tuning modules, and including this work in the literature review would be beneficial.
6. Typos:
- L028: Parameter Efficient Fine Tuning -> Parameter-Efficient Fine-Tuning
- L114: data driven -> data-driven
- L409: (RAG) Lewis et al. (2020) -> (RAG; Lewis et al., 2020)

References

[1] Zhang, J., Liu, J., & He, J. (2023). Composing parameter-efficient modules with arithmetic operation. Advances in Neural Information Processing Systems, 36, 12589-12610.

**Questions:**

1. The analysis concerning “cumulative explained variance” is based on three layers of the model. However, it is unclear how these three layers were selected and why only three layers were chosen for analysis.
2. The cumulative explained variance for singular values is calculated for the k_proj layers. What about the other layers? Did they exhibit similar patterns?

---

### Note · Authors · 2024-11-15

I have read and agree with the venue's withdrawal policy on behalf of myself and my co-authors.